Genome-wide association study on color-image-based convolutional neural networks

Liu Han-Ming 1 lhmgzjx@163.com
Liu Zhao-Fa 2
Li Zi 1
Yu Cong 1
Hu Peng-Cheng 1
Liu Qi-Feng 1
Shi Tai-Gui 1
1 School of Mathematics and Computer Science, Gannan Normal University , Ganzhou , China
2 Ganzhou Teachers College , Ganzhou , China
Uversky Vladimir
Electronic publication date: 2025 Jan 13
Publication date: 2025
Volume: 13
Electronic Location ID: e18822
Received 2024 Jul 17; Accepted 2024 Dec 16
Copyright: © 2025 Liu et al.
Copyright year: 2025
Copyright holder: Liu et al.
License: This is an open access article distributed under the terms of the Creative Commons Attribution License, which permits unrestricted use, distribution, reproduction and adaptation in any medium and for any purpose provided that it is properly attributed. For attribution, the original author(s), title, publication source (PeerJ) and either DOI or URL of the article must be cited.
License URL: https://creativecommons.org/licenses/by/4.0/

Keywords: Genome-wide association study, Color image, Convolutional neural network

Funding: National Natural Science Foundation of China 31660321 Science and Technology Program Foundation of Jiangxi Education Committee of China GJJ2201203 This work was supported by the National Natural Science Foundation of China (grant No. 31660321), and the Science and Technology Program Foundation of Jiangxi Education Committee of China (grant No. GJJ2201203). The funders had no role in study design, data collection and analysis, decision to publish, or preparation of the manuscript.

==============================
Background

Convolutional neural networks have excellent modeling abilities to complex large-scale datasets and have been applied to genomics. It requires converting genotype data to image format when employing convolutional neural networks to genome-wide association studies. Existing studies converting the data into grayscale images have shown promising. However, the grayscale image may cause the loss of information of the genotype data.

Methods

In order to make full use of the information, we proposed a new method, color-image-based convolutional neural networks, by converting the data into color images.

Results

The experiments on simulation and real data show that our method outperforms the existing methods proposed by Yue and Chen for converting data into grayscale images, in which the model accuracy is improved by an average of 7.61%, and the ratio of disease risk genes is increased by an average of 18.91%. The new method has better robustness and generalized performance.

Introduction

A single nucleotide polymorphism (SNP) is a DNA sequence polymorphism caused by a single nucleotide. The SNP located in the coding region is very likely to directly affect the structure and expression level of the protein, that is, it may associate with a disease. Therefore, a SNP-based genome-wide association study (GWAS) is an efficient ways of locating disease risk genes and of insight into the genetic mechanism of complex diseases.

Deep neural networks (DNNs) have shown to outperform traditional machine learning algorithms in many applications. For example, Badré et al. (2021) conducted a detailed comparative study of DNN by using polygenic risk scores, and found that a DNN outperforms alternative machine learning methods and established statistical algorithms. In 2012, the first DNN AlexNet (Krizhevsky, Sutskever & Hinton, 2012) was used for image classification by employing a deep convolutional neural network (CNN). Then, more DNN models have been proposed, such as P-NET (Elmarakeby et al., 2021), VGG (Simonyan & Zisserman, 2015), network in network (NIN) (Lin, Chen & Yan, 2014), Inception (Szegedy et al., 2015), ResNet (He et al., 2016), DenseNet (Huang et al., 2017), NASNet (Zoph et al., 2018), SENet (Hu, Shen & Sun, 2018) and ImageNet (Deng et al., 2009), and they are all based on CNNs (Cun et al., 1989; LeCun, Bengio & Hinton, 2015). CNN is one of the representative algorithms of deep learning (Gu et al., 2018; Ian, Bengio & Courville, 2016), which is a feedforward neural network via employing convolution operations. A CNN is mainly composed of an input layer, a combination of convolutional and pooling layers, and a fully connected multi-layer perceptual classifier. Due to its excellent modeling abilities to complex large-scale datasets, CNN has already been introduced into the bioinformatics, such as motif location (Alipanahi et al., 2015), deleterious variants prediction (Quang, Chen & Xie, 2014), gene expression inference (Chen et al., 2016) and DNA/RNA sequence binding specificities (Trabelsi, Chaabane & Ben-Hur, 2019).

The CNN was designed for image classification, that is, its input data is image format. In order to introduce it into GWAS, the existing works usually convert genotype data to grayscale images (Chen et al., 2021; Sun et al., 2019; Yue et al., 2020) to meet the input format of a CNN. For simplicity, we named these methods as gray-based conversion (GC) here. Since the gray images may lose more information of the data, we proposed a color-image-based genotype data conversion method named color-image-based conversion (CC) to improve the performance of CNN in GWAS. The experiments show that our method significantly outperforms the existing methods.

Materials and Methods

Datasets

Simulation data

To compare the performance of GC and CC on CNN used in GWAS, we employed PLINK (Purcell et al., 2007) to generate 10 genotype sets as 10 separated simulation datasets, in which each set contains 8,000 samples (4,000 cases and controls respectively), and each sample consists of 2,000 SNPs (including 200 risk SNPs). We coded the datasets according to the requirements of GC and CC firstly, then each sample of each set was arranged to a 45 × 45 matrix (the rest was filled with 0) and saved the matrices as images.

Real data

The real data was used to further verify the effectiveness of CC. We employed the bipolar disorder (BD) dataset from the Wellcome Trust Case Control Consortium (WTCCC) (Burton et al., 2007), as used Yue et al. (2020). Similar to the GC methods (Chen et al., 2021; Sun et al., 2019; Yue et al., 2020), we conducted necessary quality control and preliminary association analysis on the original data to ensure the quality of the data and reduce the runtime of the later CNN. The cutoffs of the quality control are SNP deletion rate of 5% among individuals, individual deletion rate of 5% among SNPs, minor allele frequency (MAF) of 5% and Hardy-Weinberg equilibrium (HWE) of 1%. After the quality control, we obtained 3,498 samples (1,998 cases and 1,500 controls) and each sample includes 363,451 SNPs (excluding the sex chromosomes). Then, the data was conducted preliminary association analysis by PLINK and obtained 6,492 risk SNPs when the P threshold is 0.01 and 25,220 risk SNPs when the threshold is 0.05. Here, we name the two groups of SNPs as P001 and P005, respectively. Finally, the datasets were converted into images of size of 81 × 81 for P001 and images of size of 159 × 159 for P005.

Gray-image-based conversion

The CNN was only introduced to GWAS in recent years, and there are a few literatures in this field. Sun et al.’s (2019) work encodes the 10 genotypes consisting of bases A, C, G and T with 4-bit binary and arranges the encoded data to a matrix (the rest of the matrix is filled with 0), and then saves the matrix as a single-channel grayscale bitmap. Yue et al.’s (2020) work improved the work of Sun et al. (2019) via (a) encoding the 10 genotypes as 1~10, (b) multiplying each code by 25 to make the code close to the pixel value of 0~255, and (c) arranging the chromosome codes orderly to a matrix (the rest of each row is filled with 0), and then saves the matrix as a single-channel grayscale bitmap (Fig. 1). It is different from the two works, Chen et al.’s (2021) work encodes the three genotypes AA, Aa and aa consisting of only major and minor alleles as 0, 154 and 254, and then arranges the codes to a matrix and saves it as a grayscale image, too.

Figure 1 A sample image of GC.

Chromosome codes are arranged orderly (the rest of each row is filled with 0). The grayscale bitmap has been magnified 10 times.

Color-image-based conversion

The methods of Sun et al. (2019) and Yue et al. (2020) make full use of the information of genotype data consisting of four bases. However, there are many of genotype datasets are include only major and minor alleles and thus, which leads the methods don’t work well. Furthermore, the matrices of the works of Sun et al. (2019), Yue et al. (2020) and Chen et al. (2021) can only be converted into grayscale images, which may lose information of the data. In order to reduce the loss of the information and improve the performance of the CNN in GWAS, we proposed a color-image-based genotype data conversion method, namely CC. The CC method encodes the three genotypes AA, Aa and aa consisting of major and minor alleles as (0, 0, 255), (0, 170, 0) and (85, 0, 0). Then, for each code, we consider the three groups of numbers as the red, green and blue (R, G and B) channels of a color image respectively (Table 1). Finally, for each sample, the codes of the chromosomes are arranged end to end to a three-dimension (3D) matrix of n × n × 3 (n is the length of the image side, the 3 represents the R, G and B channels, and the rest of each n × n matrix is filled with 0, Fig. 2A) and saves the 3D matrix as a color bitmap (Fig. 2B).

Table 1 CC coding.

	R	G	B	
AA	0	0	255	
Aa	0	170	0	
aa	85	0	0	

Figure 2 A sample image of CC.

(A) Chromosome codes arranged end to end (one channel, the blank at the end is filled with 0). (B) Color bitmap.

Results

To evaluate the performance of our method, this study employed the existing works of Yue et al. (2020) and Chen et al. (2021) as the benchmarks. The reason why the work of Sun et al. (2019) was not used here is because Yue et al. (2020) verified that the performance of Sun et al. (2019) is not as good as Yue et al. (2020).

Simulation experiments

The existing works construct CNNs by specifying network parameters. These construction methods are debatable, because there are many parameters (i.e., hyperparameters) in a CNN, and making it be difficult to ensure that the parameter settings do not produce preferences. In order to avoid the effects as much as possible, the Bayesian method (Krizhevsky & Hinton, 2009) was used to optimize the hyperparameters in this study, and the parameters to be optimized are shown in Table 2. We divided each imaged dataset into training, validation and testing sets according to the ratio of 6:2:2 firstly, then constructed three CNN classification models for CC and two GCs by Bayesian optimization method, and last used the optimized models to predict whether the samples are cases or controls one by one to test the models. The input sizes of the models of CC and GC were set to 45 × 45 × 3 and 45 × 45 × 1, respectively. Each model contains three convolutional layers with the same optimizing filter size and three pooling layers. Following each convolutional layer, a maximum pooling layer (stride = 2) was added. Moreover, we employed a dropout strategy (Srivastava et al., 2014) on the last convolutional layer to reduce overfitting. Furthermore, the other adjustable parameters were set to default values. After a maximum of 60 epochs, the testing scores of the models on 10 separated sets were obtained as shown in Table 3.

Table 2 Parameters to be optimized.

Parameter	Value	
Filter size	3, 5, 7	
Learning rate	10−3~1	
Momentum	0.9~0.98	
L2 regularization	10−10~10−2	

Table 3 Average scores in simulation sets*.

Method	Accuracy	Recall	Precise	F1	
GC (Yue)	0.500 (0.00)	0.500 (0.00)	0.700 (0.00)	0.583 (0.00)	
GC (Chen)	0.966 (8.23 × 10−3)	0.965 (9.17 × 10−3)	0.967 (8.52 × 10−3)	0.966 (8.29 × 10−3)	
CC (Ours)	0.973 (9.48 × 10−3)	0.975 (1.04 × 10−2)	0.972 (1.05 × 10−2)	0.973 (9.49 × 10−3)	
Note:

* The real numbers in the parentheses are the standard deviations.

Real experiments

Optimizing CNN classification model

Just like the simulation experiments, we optimized the CNN classification models for the data groups of P001 and P005 by predicting a sample is a case or control, too. Except for setting the input size of the model according to “Materials and Methods”, the optimized parameters, as well as the ratio of the training, validation and testing sets, are similar to the simulation experiments. The testing scores after a maximum of 60 training epochs on the models are shown in Table 4.

Table 4 Scores of real sets.

Set	Method	Accuracy	Recall	Precise	F1	
P001	GC (Yue)	0.843	0.842	0.780	0.810	
GC (Chen)	0.843	0.778	0.887	0.829	
CC (Ours)	0.860	0.869	0.793	0.829	
P005	GC (Yue)	0.710	0.675	0.623	0.648	
GC (Chen)	0.693	0.684	0.527	0.595	
CC (Ours)	0.794	0.791	0.707	0.746	

Screening of risk genes

Similar to Yue et al. (2020), this study used gradient-weighted class activation mapping (Grad-CAM) (Selvaraju et al., 2017) to get the importance of an SNP used to indicate whether the SNP is risk or not.

We obtained a matrix after accumulating the two-dimensional Grad-CAM map matrices of the test samples whose outputs of the classification are cases. Then the matrix was normalized as a weight matrix whose elements represent the contribution of pixels of the sample image. Since each pixel of the image in this study represents an SNP, we let the weight of each SNP be the risk probability of the gene involving the SNP of BD. As the Yue et al.’s (2020) work, we took out the top of 5% SNPs as the risk SNPs and queried the genes on NCBI (http://www.ncbi.nlm.nih.gov/) (Brown et al., 2015; Sherry et al., 2001) to match them (the SNPs without matched genes were discarded). If a gene contains several SNPs, but not all SNPs lie in the top of 5%, then only the SNPs located in the top of 5% are considered as a risk. Finally, the group P001 was screened a total of 407 SNPs involved in 180 genes and, the group P005 was screened a total of 1,375 SNPs involved 419 genes. By querying the DisGeNET (https://www.disgenet.org/) database (Bauer-Mehren et al., 2010; Piñero et al., 2016), 87 of the 180 genes in P001 are risk genes, and 199 of the 419 genes in P005 are risk genes. The numbers of the screened SNPs and genes of the two groups are shown in Table 5.

Table 5 The numbers of screened SNPs and genes.

Set	Method	Associated SNP*	Associated gene*	Ratio	
P001	GC (Yue)	112	41	0.366	
GC (Chen)	83	35	0.422	
CC (Ours)	62	31	0.500	
P005	GC (Yue)	290	79	0.272	
GC (Chen)	327	91	0.278	
CC (Ours)	325	99	0.305	
Note:

* The two sets of numbers are the counts of SNPs associated with BD among the screened SNPs and the genes containing the associated SNPs, respectively.

The results of the simulation experiments show that the testing accuracy, recall rate, precision and F1 scores of the methods of GC and CC are almost the same (Table 3). It indicates that the performance of CNN in a noise-free GWAS is hardly affected by the ways of converting genotypes into images. Nonetheless, the results of real experiments in Table 4 show that the CC testing accuracy is average increased by 2.02% when the pre-screening P threshold is 0.01, and especially the accuracy is average increased by 13.19% when the threshold is 0.05. That is, the performance of our method CC is moderate in the noise-free simulation data, but is better in the P001 real data with a little noise, and is excellent in the P005 real data with more noise. Thus, CC has better robustness. Furthermore, the numbers of screened associated genes in Table 5 show that the ratio of risk genes of CC is average improved by 26.91% when the P threshold is 0.01 and is average improved by 10.91% when the threshold is 0.05. The ratios suggest that the GC methods tend to the SNPs involved in a single gene while CC method is the opposite. It means the color-image-based converting method CC has better generalized performance.

Discussion

The experiments suggest that although the color-image-based CC method has moderate performance in noiseless simulation data, it is significantly improved in noisy real data. The CC method has better robustness and generalized performance. Why can the CC method improve the performance of CNN in GWAS? Table 1 show that CC method essentially encodes genotypes AA, Aa and aa as blue, green and red pixels of the image. Suppose the sequence of a sample is AA, aA, Aa, double aa, Aa and double AA, then the images produced by GCs and CC are shown in Fig. 3. As shown in the figure, because Sun et al. (2019) uses 2-bit binary to encode a pixel, its grayscale difference becomes very small, and leads to the fact that its pixels are almost fused together and difficult to be distinguished (Fig. 3A); Yue et al. (2020) encodes the genotypes with decimal 1~10 and multiplies the codes by 25. It extends the grayscale difference and leads its image resolution is much better than that of Sun et al. (2019) (Fig. 3B); Chen et al. (2021) encodes genotypes AA, Aa and aa as 0, 154 and 254 respectively, and further extends the grayscale difference and makes its image resolution be greatly improved compared with Sun et al. (2019) and Yue et al. (2020) (Fig. 3C). Although the existing methods have been improved one by one, there is still a gap in resolution compared with CC method. It is because (a) the resolution of a color image is naturally higher than that of a grayscale image; (b) Chen et al. (2021) with the highest resolution, codes both genotype AA and blank (i.e., the rest of the matrix) as 0, will lead CNN to treat a genotype AA as blank (or a blank as genotype AA) later and reduce the resolution actually. The higher resolution an image is, the better classification CCN has (Chevalier et al., 2015; Liu et al., 2016). Moreover, GC methods put the codes of each chromosome at the beginning of the image rows (Fig. 1), lead to mix blanks with pixels and aggravate the interference to the image.

Figure 3 Image comparison between GC and CC.

The images generated by the frameworks of (A) Sun et al.’s (2019) GC, (B) Yue et al.’s (2020) GC, (C) Chen et al.’s (2021) GC and (D) our method. The “0”es are filled in as the rest of the matrices.

When a color image is inputted to a CNN, the CNN employs the R, G and B layers as three separated channel inputs and connects all filters of the three channels as its output (Fig. 4A). However, a grayscale image was overlapped before being inputted and is processed as a single-channel in CNN (Fig. 4B). Therefore, although both a color and grayscale image will become one-channel before being outputted, the CNN procedure of a color image is three separated channels, which can better weaken the interference between pixels than a grayscale image. The interference is little effect on a common image with rich pixels, but it cannot be ignored for a genotypic image with only three kinds of pixels. Mathematically, for CC method, the three kinds of genotypes AA, Aa and aa will be inputted into three different CNN channels, while for GC method, they will be inputted into the same channel. CNN will train a suitable set of weights for each channel, which means that CC method in CNN will obtain three different sets of weights, enabling it to better distinguish between the three kinds of genotypes. In contrast, the GC methods in CNN are only assigned one set of weights to the three kinds of genotypes, naturally reducing its ability to distinguish between different genotypes. Of course, for noiseless simulation data, the low discriminative ability of GCs has little limitation. That is why the CC method did not show outstanding performance in simulation experiments, but it was significantly better than the GC methods in noisy real experiments. Thus, employing color images is a good choice to CNN in GWAS.

Figure 4 CNN.

(A) Multi-channel CNN of color image. (B) Single-channel of grayscale image.

Of course, compared to Sun et al. (2019) and Yue et al. (2020) of 10 genotypes, CC has only three. It may theoretically lower its performance, but the influence is so little that it can be ignored. This is because (a) base pairs A-T and G-C are same in nature; (b) base pairs other than A-T and G-C account for a low proportion in genomes. This is approved by Chen et al. (2021) that encodes only three genotypes but has better performance than the existing methods.

In terms of application, CC method can be easily used for genotype data in the format of major-minor allele pairs or for converting other format data into this format. An existing work provides a format conversion tool for common genotype data (Liu et al., 2020). Although CC method might achieve better performance, it requires more storage and memory spaces compared to GC methods, which is a research direction that needs to be addressed in the future.

Conclusions

A CNN is constructed by mimicking the visual perception mechanism of living organisms. The shared convolutional kernel parameters of its hidden layers and the sparsity of inter layer connections enable CNN to learn grid-like topology features, such as pixels, with low computational complexity and stable performance. The genotype data faced by GWAS, similar to pixels, has an extremely high-dimensional data. It is an interesting work to explore the application of CNN in GWAS by leveraging its advantages in extremely high-dimensional data mining. Due to CNN’s expertise in grid-like topology features, it is necessary to convert genotype data into the topology before introducing CNN to GWAS. The existing works that convert genotype data into grid-like topology by grayscale images, which may lead to information loss, while converting to color image topology could save more information from the original data. Our method converting genotype data to color images in this study can improve the testing and screening accuracy of CNN in GWAS compared with the existing works. It is worth to further study to employ it to the other biomedical field in the future works.

Supplemental Information

Supplemental Information 1 Supplementary Tables.

Supplemental Information 2 Raw data.

Additional Information and Declarations

Competing Interests

Author Contributions

Data Availability

The authors declare that they have no competing interests.

Han-Ming Liu conceived and designed the experiments, analyzed the data, authored or reviewed drafts of the article, and approved the final draft.

Zhao-Fa Liu performed the experiments, analyzed the data, prepared figures and/or tables, authored or reviewed drafts of the article, and approved the final draft.

Zi Li performed the experiments, analyzed the data, prepared figures and/or tables, authored or reviewed drafts of the article, and approved the final draft.

Cong Yu analyzed the data, prepared figures and/or tables, and approved the final draft.

Peng-Cheng Hu analyzed the data, prepared figures and/or tables, and approved the final draft.

Qi-Feng Liu analyzed the data, prepared figures and/or tables, and approved the final draft.

Tai-Gui Shi analyzed the data, prepared figures and/or tables, and approved the final draft.

The following information was supplied regarding data availability:

The raw sequence data (500K Affymetrix chip data) are available at the Wellcome Trust Case Control Consortium (WTCCC): https://www.wtccc.org.uk/info/access_to_data_samples.html.

The codes for this study are available at GitHub: https://github.com/spvm2000/CC-CNN/tree/main.

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
