# Peer review of "Genome-wide association study on color-image-based convolutional neural networks"

_PeerJ, doi:10.7717/peerj.18822_

## Round 0.1 · original submission · Major Revisions

Please address the concerns of all the reviewers and revise manuscript accordingly.

Reviewer 1 ·

Basic reporting

In the paper titled "Genome-wide association study on color-image-based convolutional neural networks,” the authors propose a novel method for genome-wide association studies (GWAS) by converting genotype data into color images and utilizing color-image-based convolutional neural networks (CNNs). While existing studies have primarily converted genotype data into grayscale images, the authors argue that color images preserve more information, leading to higher accuracy rates. The method was tested on both simulated and real-world datasets, and the results demonstrate that the color-image-based conversion (CC) outperforms grayscale-based methods (GC).

Experimental design

This study introduces a new approach to processing genotype data. The potential to preserve and process more information using color images could enhance the applicability of deep learning techniques in genomic research. This innovation could have a substantial impact on future biomedical research and the identification of disease-risk genes.
Strengths:
• The authors effectively demonstrate the accuracy and robustness of their proposed method using both simulated and real data, supporting the method's practical applicability.
• The finding that color-image-based conversion performs better, particularly on noisy datasets, suggests that the method could be generalized to a broader range of data.
• The structure and presentation of the paper are clear and effectively convey the subject matter and the supporting results.
Weaknesses and Suggestions for Improvement:
• Abstract: The results section is incomplete and lacks specifics regarding existing methods (lines 26-27).
• Experimental Diversity: Although the datasets used in the paper are sufficient to demonstrate the method's effectiveness, the simulation dataset includes a small sample size (lines 62-68).
• Methodological Clarity: The Methods section should provide more detailed explanations, particularly by adding mathematical explanations.
• In-Depth Discussion of Results: While the authors discuss the results adequately, providing more information on the potential limitations of the method would be beneficial. Specifically, addressing the applicability of color-image-based conversion to different types of genomic data and its potential limitations could add value.
• Language: The language should be improved, especially in lines 117-120.

Validity of the findings

This study presents a significant innovation in processing genomic data and could make a valuable contribution to the field of bioinformatics. The paper could be further strengthened with minor improvements and additional experiments. I recommend the publication of this paper after the authors address these suggestions.

Reviewer 2 ·

Basic reporting

The study introduces a new method for converting genotype data into color images, enhancing the use of convolutional neural networks (CNNs) in genome-wide association studies (GWAS).
Both simulated and real genotype data were used to evaluate the performance of the CC method.

Simulation datasets included 10 genotype sets with 4,000 samples each, while real data was sourced from the Wellcome Trust Cast Control Consortium (WTCCC) bipolar disorder dataset.

The study compares the CC method with existing grayscale conversion methods, showing that the CC method significantly improves CNN performance in identifying disease risk genes.

Specifically, the model accuracy increased by an average of 7.61%, and the identification of disease risk genes improved by an average of 18.91%.

The CNN models were optimized using the Bayesian method to fine-tune hyperparameters, avoiding biases in parameter settings.

The study ensured that the models for both CC and grayscale conversion (GC) had similar architectures, with differences mainly in the input image format and color channels.

The results indicate that the CC method not only enhances accuracy but also demonstrates better robustness and generalization across different datasets, making it a promising approach for future GWAS applications involving CNNs.

Experimental design

comparing a new color-image-based genotype data conversion method (CC) with existing grayscale conversion methods (GC) in genome-wide association studies (GWAS). Using both simulated and real genotype datasets, CNN models were trained and optimized using Bayesian methods to minimize biases. The performance of CC and GC was then evaluated based on model accuracy and the identification of disease risk genes, with CC showing superior results in both robustness and generalization.

Validity of the findings

The use of a Bayesian method to optimize CNN hyperparameters minimizes potential biases, strengthening the reliability of the comparisons between the color-image-based (CC) and grayscale conversion (GC) methods.The improved model accuracy and increased identification of disease risk genes with CC were consistent across multiple datasets, supporting the robustness of the findings.The experiment's design, which included controlled testing on separated datasets, enhances the internal validity of the results, reducing the likelihood of random errors.The consistency of the CC method's superior performance across various scenarios indicates strong external validity, suggesting that these findings could be generalized to other genome-wide association studies.

Additional comments

The color-image-based conversion (CC) method may require more computational resources and processing time compared to grayscale conversion due to the increased data dimensionality (three color channels versus one). Encoding genotype data into color images could introduce noise or artifacts if not carefully managed, potentially affecting the accuracy of the CNN models. More implementation details required that is missing in manuscript. As genome-wide association studies scale to larger datasets, the CC method might face challenges in handling the increased volume of data efficiently. The CC method may be less effective or require significant adaptation when applied to other types of genomic data that do not translate easily into color image formats.

Reviewer 3 ·

Basic reporting

The article is well-written, with some minor mistakes (e.g. sentence on 119 is not finished). Relevant literature is referenced, but with a very limited scope. For example there exist many types of neural networks for this type of task that do not use image-based CNNs:

Elmarakeby, H. A., Hwang, J., Arafeh, R., Crowdis, J., Gang, S., Liu, D., AlDubayan, S. H., Salari, K., Kregel, S., Richter, C., Arnoff, T. E., Park, J., Hahn, W. C., & Van Allen, E. M. (2021). Biologically informed deep neural network for prostate cancer discovery. Nature, 598(7880), 348–352. https://doi.org/10.1038/s41586-021-03922-4

Badré, A., Zhang, L., Muchero, W., Reynolds, J. C., & Pan, C. (2021). Deep neural network improves the estimation of polygenic risk scores for breast cancer. Journal of Human Genetics, 66(4), 359-369.

There are some references broken on lines:93 94, 109, 112, 113, 114, 135 and the formatting in the bibliography should be improved. The figures would benefit from a more elaborate captions. Figure 4 could describe the CNN better, it is now a bit confusing. Maybe avoid using the same dashed box for the input and the CNN. More details about the network would be appreciated (e.g., number of layers, type of activation). The first part of the discussion, where you describe the results, should be still part of the result section.

The paper is self-contained and the authors test a clear hypothesis.

Experimental design

The article fits the scope of the journal.

The authors try to improve a specific type of CNN by adding additional channels. The hypothesis is clear and the authors succeed in getting better results than the original architecture. The work would be more meaningful if different types of networks were evaluated. For example comparing the network with visible neural networks or regular feedforward neural networks.

Questions regarding the experimental design:

If you perform GWAS to select SNPs, interpreting these SNPs and checking if they are relevant is double dipping. All SNPs in your network are already important for the disease since you excluded SNPs not below a p-value threshold. Maybe plot the p-values vs the importance values and inspect deviations. Scientifically more interesting would be to test if a linear model with all these SNPs perform worse than your neural network. This would be relatively easy to do and will tell you if a complex model such as a CNN does do actually better than polygenic risk score (from your mini GWAS).

In your GWAS to select SNPs you should not have used subjects from the test set. If you use the test set for feature selection you will get bias results.

Why not use early stopping? (might give better performance). Maybe you are already overfitting after 60 epochs. Please provide which hyperparameters (more common word than superparameters) you optimize on your validation set.

Please provide the standard deviation or 95% confidence interval for table 1.

Would be good to release the code on GitHub or Zenodo.

Validity of the findings

The code is necessary for replication.

"The existing works that convert genotype data into grid-like topology by grayscale images, which may lead to information loss, while converting to color image topology could save more information from the original data."

1. Genotype data with SNPs from different locations in the genome do not have a local structure. CNNs are made with the intention on combining local information, although you can use them for this type of data, it seems not the most logical choice to me. It would be good to motivate this choice.

2. Is there actually an information loss when converting the data to greyscale image? I suspect that the increased number of parameters for the model that you get when using more channels and the better format is responsible for the improvements.

line 228 "This was verified by the GWAS experiments in this study,". I don't see how the GWAS experiment verified this.

It would be good if the authors revisit the conclusion and clarify the conclusion.

---

## Round 0.2 · Minor Revisions

Please address remaining issues pointed by the reviewer and revise manuscript acordingly.

Reviewer 2 ·

Basic reporting

The revised writing show well-documented and provides a clear rationale for the proposed color-image-based CNN method.

Experimental design

The experimental setup is robust, with result and real data supporting the method's effectiveness.

Validity of the findings

The improved accuracy and detection of disease risk genes validate the superiority of the proposed method over grayscale image-based approaches

Additional comments

Ensure that all references are correctly cited in the text and listed in the bibliography. Cross-check for missing or incorrectly formatted citations. Clarify sentences to enhance readability and ensure proper grammar throughout the paper. Ensure all abbreviations (e.g., CNN for convolutional neural networks) are defined at first use and consistently applied throughout the paper. Verify that all figures are correctly numbered and referred to in the text in sequential order. Confirm that figure captions are clear and descriptive, and match the figure content. Ensure uniformity in font styles, headings, and overall formatting as per the journal's guidelines.

---

## Round 0.3 · accepted · Accept

All remaining issues were addressed and revied manuscript is acceptable now.